# Safety, Efficacy and High-Quality Standards of Gastrointestinal Endoscopy Procedures in Personalized Sedoanalgesia Managed by the Gastroenterologist: A Retrospective Study

**DOI:** 10.3390/jpm12071171

**Published:** 2022-07-19

**Authors:** Marina Rizzi, Francesco Panzera, Demetrio Panzera, Berardino D’Ascoli

**Affiliations:** 1Interventional Gastroenterology Unit, Madonna delle Grazie Hospital in Matera, 75100 Matera, Italy; francescopanzera@hotmail.com (F.P.); bdascoli@libero.it (B.D.); 2Division of Cardiac Anesthesia, Department of Intensive Care, Vito Fazzi Hospital, 73100 Lecce, Italy; demetrio.panzera@libero.it

**Keywords:** sedoanalgesia, GI endoscopy, fentanyl, midazolam, colonoscopy, ADR, cecal intubation

## Abstract

Performing GI endoscopy under sedoanalgesia improves the quality-indices of the examination, in particular for cecal intubation and adenoma detection rates during colonoscopy. The implementation of procedural sedoanalgesia in GI endoscopy is also strongly recommended by the guidelines of the major international scientific societies. Nevertheless, there are regional barriers that prevent the widespread adoption of this good practice. A retrospective monocentric analytic study was performed on a cohort of 529 patients who underwent EGDS/Colonoscopy in sedoanalgesia, with personalized dosage of Fentanyl and Midazolam. ASA class, age and weight were collected for each patient. The vital parameters were recorded during, pre- and post-procedure. The rates of cecal intubation and of procedure-related complications were entered. The VAS scale was used to evaluate the efficacy of sedoanalgesia, and the Aldrete score was used for patient discharge criteria. No clinically significant differences were found between vital signs pre- and post-procedure. Both anesthesia and endoscopic-related complications occurring were few and successfully managed. At the end of examination, both the mean Aldrete score (89.56), and the VAS score (<4 in 99.1%) were suitable for discharge. For the colonoscopies, the cumulative adenoma detection rate (25%) and the cecal intubation rate in the general group (98%) and in the colorectal cancer screening group (100%) were satisfying. Pain control management is an ethical and medical issue aimed at increasing both patient compliance and the quality of the procedures. The findings of this work underscore that in selected patients personalized sedoanalgesia in GI endoscopy can be safely managed by gastroenterologists.

## 1. Introduction

The demand for gastrointestinal (GI) endoscopic examinations has been increasing all over the world, given the growing need for early diagnosis of oncological and inflammatory pathologies. There is also a growing demand by both doctors and patients to reduce the anxiety and pain related to these invasive procedures, to make them tolerable and, where necessary, readily repeatable without excessive anxiousness. Out of all the invasive procedures, GI endoscopic procedures are, in fact, considered among those which result in the most patient discomfort, embarrassment, fear and pain [1], which can affect their compliance with the prescribed instructions, with serious diagnostic delays of pathologies, such as cancer and chronic inflammatory bowel diseases, for which diagnostic timing is crucial [2]. Therefore, the management of procedural pain appears to be an ethical imperative aimed at respecting the dignity of the person, improving diagnostic timing and quality of care. Performing a GI endoscopic examination under sedation improves the quality of the examination itself, in particular the colonoscopy quality indices such as the adenoma detection rate (ADR) and the cecal intubation rate [3]. The implementation of procedural sedoanalgesia during a screening colonoscopy is therefore strongly recommended by the ESGE [4]. There are different pharmaceutical drugs that can be used for procedural sedoanalgesia, and the most frequently used intravenous formulations are opioids, sedatives and/or propofol [5]. When contemplating the use of propofol for sedoanalgesia, which is considered highly effective [6], there are often different local guidelines and/or regulations to take into consideration, namely the fact that in some countries it cannot be administrated by a gastroenterologist or nurse but only by an anesthetist [7]; in addition, with propofol, the pre-, intra- and post-procedural management can be more complex and time-consuming, requiring different skills [8].

When considering the use of sedatives and opioids, the dosage of said pharmaceutical drugs must be correctly balanced on the basis of the comorbidities, weight, age and patient response to obtain a good analgesic and sedative result and minimize the possibility of side effects [9]. Hence, the doctor—not necessarily an anesthetist—who uses these pharmaceutical drugs must know how to manage them, immediately recognize and treat any adverse events and use the antagonists with dexterity. Finally, the need for procedural sedoanalgesia must be in line with the organizational structure of the hospitals, which cannot always guarantee the presence of a dedicated anesthetist in the endoscopy room, and with the needs of health economics aimed at optimizing staff and resources. Despite, and due to all the premises listed thus far, the use of procedural sedoanalgesia in GI Endoscopy differs greatly between countries, as well as between local hospitals, with regard to the figures involved (anesthetist or gastroenterologist), the pharmaceutical drugs used and patient monitoring pre-, intra- and post-examination; scientific studies are very contradictory with regard to the indications on the safety and efficacy of the pharmaceutical drugs used in this setting [10]. 

For all the aforementioned reasons, the Interventional Gastroenterology Unit, Madonna delle Grazie Hospital in Matera has since 2019 implemented the use of sedoanalgesia in GI endoscopy in selected patients with a patient-personalized dosage of sedative (midazolam) and opioids (fentanyl) pharmaceutical drugs that are conducted by expert gastroenterologists trained for emergencies and the use of said drugs. 

The primary objective of this retrospective study is to verify the safety and efficacy of GI endoscopy procedural sedoanalgesia conducted by a gastroenterologist in selected patients with patient-personalized dosages of sedative (midazolam) and opioid (fentanyl) pharmaceutical drugs. The secondary objective is evaluating the quality of the colonoscopies conducted with sedoanalgesia, namely the cecal intubation rate, ADR and polyp detection rate (PDR) [11]. 

## 2. Materials and Methods

A retrospective analytical monocentric study was carried out on a cohort of 529 patients who underwent an esophagogastroduodenoscopy (EGDS) and/or colonoscopy with sedoanalgesia between February 2019 and February 2020. The endoscopic examinations with sedoanalgesia were performed by a team of three experienced endoscopists with comparable operative digestive endoscopy and sedative drug management skill levels. Prior to the endoscopic examination, a detailed anamnesis was collected for each patient, including age, weight, relevant co-morbidities, drug allergies and prior abdominal and/or proctological surgery, to assess the correct ASA class for each patient (Table 1). The exclusion criteria for this study were an age of <16 or >85 years, ASA class > 3, pregnancy, psychiatric disorders and known allergies to sedoanalgesic drugs. First, this pre-procedure testing has allowed us to select the patients to place under sedoanalgesia with fentanyl and midazolam managed by the gastroenterologist, more specifically those belonging to ASA class 1 to 3 and, second, to personalize the dosage of the aforementioned drugs for each patient based on age, comorbidity, weight and ASA score.

Each patient was fully informed with regard to the examination to be performed and the sedation and sedoanalgesia options chosen, and each patient signed an informed consent for the endoscopic examination and the sedation, which was countersigned by the endoscopist gastroenterologist. The two pharmaceutical drugs used were a benzodiazepine (midazolam) at a dosage of 0.025 mg/kg and an opioid (fentanyl) at two different dosage levels: 100 mcg in patients weighing over 50 kg and aged 70 years or less and 50 mcg in patients weighing less than 50 kg and aged 70 years or more. Pharmaceutical drug dosages were reduced in cases of significant co-morbidity, for a correct personalized sedoanalgesia. The pharmaceutical drugs were administrated using the following protocol: an infusion of Fentanyl followed by Midazolam, careful titrating the effect of the two drugs based on the subject’s response. The following vital parameters of each patient were measured one time both pre- and post-procedure: heart rate (HR); peripheral O_2_ saturation with pulse oximetry (SpO_2_); non-invasive blood pressure monitoring (BP). During the examination, each patient was connected to a continuous multi-parameter monitor to measure SpO_2_ and HR. A peripheral venous access was placed on each patient prior to the examination. The administration of O_2_ via nasal cannula was provided only for persistent reductions in SpO_2_ < 90% for over 60 s. The Ramsey score [12] was used to assess the depth of sedation, considering safety levels from 1 to 4 (Table 2), with a target level of 3. 

All the data listed thus far were recorded on a specific form. At the end of each examination, each patient received a report with a detailed description of the outcome, including every complication and subsequent management of said complication. In the case of polyps, each one was excised and histologically evaluated. With regard to the quality indices of the examination, for colonoscopies we calculated the adenoma detection rate (ADR), the adenoma detection index (ADI), the polyp detection rate (PDR), the polyp detection index (PDI) [13] and the cecal intubation rate on the general patient population, on the patient population divided by gender and the patient population that underwent a screening colonoscopy through the regional program to screen for colorectal cancer. After the procedures, each patient was safely discharged if his Aldrete [14] score (Table 3) was higher than 8. The VAS scale, a psychometric response scale largely used as measurement tool for subjective attitudes that cannot be directly measured [15] (Figure 1), was used at the end of the exams to check the grade of pain felt before the discharge. When responding to the VAS item, patients specified their level of post procedure analgesia by indicating a position along a continuous line between 0 and 10, in which 0 corresponds to “absence of pain”, and 10 to “the worst pain ever felt”. The VAS scale was used to evaluate the efficacy of the sedoanalgesia: a score of less than 4 was deemed as a sufficient analgesic effect.

In this study, the following findings were defined as “anesthesiologic complications” or “anesthesia related complication”: hypoventilation, respiratory depression, apnea, hypotension, bradycardia and mortality [16]. More in detail: persistent reductions in SpO_2_ < 90% for over 60 s, and/or decreased level of Ramsey scale < 4, and/or HR value < 50 bpm or >100 bpm for over 60 s, and/or mean BP value > 50 mmhg or <50 mmhg of pre-sedation level were viewed as complications.

### Statistical Analyses

The categorical variables are shown by their absolute and relative frequencies. The independence of the categorical variables was verified via the Pearson’s chi-square test, or the Fisher exact tests. The normal distribution of continuous variables was verified via the Shapiro–Wilk test. The non-normally distributed continuous variables were reported by their median and interquartile range, whereas the normally distributed variables were shown through their mean and standard deviation. In the descriptive table (Table 4), the difference between the means of the groups of ASA was verified utilizing the ANOVA test. The adjusted ORs were calculated through logistic regression. Linear regression was used to assess the effect of midazolam and fentanyl on the Aldrete score. The ANOVA test—adjusted by age, weight and sex—was used to test the average difference of the midazolam and fentanyl dosages among the ASA groups (Table 4). The statistical software used for the analysis was the R version 4.1.2 (1 November 2021). 

## 3. Results

Out of 529 patients, 279 were females and 250 males. The mean age of the patients who underwent the endoscopic examination was 52.9 years (max 84, min 16, sd: 14.38) and the average body weight was 72.3 kg (max 130, minimum 45, sd: 14.13). Out of 529 patients, 443 underwent a colonoscopy, 81 underwent an EGDS and 5 both exams concurrently; 64.5% of the patients did not have prior abdominal and/or perianal surgery. A total of 92.6% of the cases reported no prior drug allergies. With regard to the provenance of these patients undergoing endoscopic examination, the vast majority were outpatient subjects (96.6%) with medical prescriptions made by a general practitioner or other specialist, while for the remaining 3.4% the requests came from hospital specialists during their hospitalization. The number of patients from the colorectal cancer screening program was 72 (Table 5). The patients were divided into the following ASA risk classes according to their prior and/or current pathologies (Figure 2, Table 4).

As for the data regarding the sedoanalgesia, the patients were administered fentanyl in 100% of cases, and midazolam was added in 99% of these cases. The mean dosage of midazolam was 2.1 mg (max: 3.5 min 1.5, sd: 0.37); the mean dosage of fentanyl was 83.93 mcg iv (max 125, min 25, sd: 23.52). The mean dosages of midazolam were not different among the ASA classes, whereas the mean dosages of fentanyl differed among the ASA classes (Table 4).

The ASA class was not associated with the development of anesthesia-related complications (*p*-value > 0.9, Table 6). With regard to the measurable vital parameters (SpO_2_, BP, HR) pre- and post-procedure, the results are as follows: the average SpO_2_ pre-procedure was 97.4% (max 100, min 88, sd: 1.83), and the average SpO_2_ post-procedure was 96.31% (max 100, min 90, sd: 2.18). The average systolic BP pre-procedure was 128.6 (max 220, min 85, sd: 19.48), and average diastolic BP pre-procedure was 77.15 (max 122, min 50, sd: 10.19). The average systolic BP post-procedure was 116.49 (max. 200, min 80, sd: 15.75), and the average diastolic BP post-procedure was 71.9 (max 130, min 50, sd: 9.74). The average HR pre-procedure was 79.6 bpm (max 140, min 40, sd:14.9); post-procedure it was 74.9 bpm (max 128 min 32, sd:11.86). With regard to mean pre/post procedure values of the SpO_2_, BP and HR, no clinically significant differences were reported. Anesthesiologic complications occurred in 4 out of 529 cases (0.8%) and involved episodes of hypotension (mean pre-sedation level BP value < 50 mmhg), contextual bradycardia (HR < 50) and reduction in SpO_2_ values < 90% lasting over 60 s; these patients were treated with Trendelenburg positioning, an IV bolus infusion of 0.9% saline solution and administration of atropine. Only one out of the four adverse events cases required the hydration of the patient with an injection of a benzodiazepine antagonist (flumazenil 0.5 mg) to contrast the excessive sedation depth measurable as a five with the Ramsey scale. The vital signs normalized immediately after the described treatment in all the aforementioned patients. Only 4% of the patients required O_2_ therapy during the endoscopic procedure due to the reduction of SpO_2_ values < 90% for over 60 s. No cases of opioid antagonist (naloxone) use were reported. No case required assisted ventilation and/or the use of supraglottic devices or orotracheal intubation. An endoscopic complication occurred in a single case: an event of post-polypectomy bleeding during a colonoscopy, which was managed with injective, mechanical and thermal combined hemostatic techniques (injection of diluted adrenaline plus argon plasma coagulation and endoclip placement), with no further endoscopic procedures required due to the absence of any signs of bleeding in the long term.

The odds ratio (OR) of anesthesia-related complications (adjusted by weight, fentanyl dosage, age and sex) increased by 52% when the midazolam dosage increased by 0.5 mg, but the (OR = 1.52) was not significantly different from 1 (*p*-value = 0.5429). Furthermore, when the midazolam dosage was increased by 0.5 mg, the mean value of the Aldrete score (adjusted by weight, fentanyl dosage, age and sex) decreased by 0.14. This result was statistically different from 0 (*p*-value ≤ 0.001). 

With regard to the data that refers to the quality of the endoscopic examinations, colonoscopies in particular, the cumulative ADR was 25%, (22% in the female group and 28% in the male group). The adenoma detection index was 37%, the polyp detection rate (PDR) was 31% and the PDI (polyp detection index) was 48%. The cecal intubation rate was 98% in the general colonoscopy group and 100% in the colorectal cancer screening colonoscopy group. It should be noted that the Boston bowel preparation score (BBPs) was not reported in the study and that an incomplete/inadequate intestinal cleansing, up to 30% reported in the literature, reduces the ADR and cecal intubation rates [17,18] (Table 7). The mean Aldrete score upon discharge was 9.56 (max 10, min 7), and in 99.1% of cases the VAS scale upon discharge showed values of <4.

## 4. Discussion 

EGDSs and colonoscopies are endoscopic procedures that are very common in clinical practice. Given their growing use in the field of diagnostic and therapeutic digestive endoscopy, the demand for these exams is constantly increasing [19]. However, these are invasive tests, often associated with discomfort, anxiety, stress and pain on the part of the patient. Sedoanalgesia in GI endoscopy ensures greater examination tolerability on the part of the patient and allows the gastroenterologist to greatly increase the quality of the examination performed. The use of sedoanalgesia for GI endoscopic procedures is strongly recommended by various guidelines issued by national and international scientific societies [20]. The sedation of patients for endoscopic examinations is now common practice in the United Kingdom, the United States and Canada. In the USA, more than 98% of the colonoscopies are performed using mild to moderate sedation, mainly by non-anesthetists, and all ERCP and ultrasound endoscopies (EUS) are performed under sedation, often using propofol [21]. In clinical practice, the management of the sedation is usually carried out directly by the gastroenterologist, except in cases of complex patients such as non-collaborating subjects, pediatric patients, very elderly patients, morbidly obese patients, pregnant women or subjects suffering from serious pathologies (cardiovascular, respiratory, hepato-renal, neurological and/or endocrine-metabolic issues) in which the management of the sedation requires the presence of an anesthesia specialist [22,23,24,25,26]. In the most serious and compromised cases, the endoscopic examination must be carried out in the operating room [27]. Furthermore, the widespread shortage of anesthetists in many countries, such as in Italy, worsened during COVID-19 pandemic [28] and has resulted in other specialists carrying out the procedures of sedoanalgesia to guarantee the greatest comfort possible to the patient undergoing the examinations. In the Italian reality, despite the indications, sedoanalgesia for GI endoscopic procedures is not yet a consolidated practice and is often considered as just a possible option [29,30]. In recent years, by virtue of the developments in the pharmacological field, the improvement of monitoring possibilities during the procedure and the cultural growth of the patients, it is becoming an increasingly widespread practice, though the approach to it is not standardized. It varies according to the institution, the organizational structure, the resources invested, the training of the health professionals involved, the complexity of the patient and the exam to be performed. In this retrospective study, we evaluated the efficacy and the safety of sedoanalgesia with fentanyl/midazolam performed by the gastroenterologist during GI endoscopy procedures in selected patients and carried out in a single first level hospital centre. In this work, the analysis of the vital parameters from the dataset of patients who underwent an endoscopy with sedoanalgesia showed that there were no clinically significant differences between the average values of BP, HR and SpO_2_ at the beginning of the procedure and those recorded at the end of the procedure, denoting the safety of sedoanalgesia in this setting. As for anesthesia-related complications, these occurred in 0.8% (4 cases out of 529), while the literature reports adverse events during sedoanalgesia with benzodiazepine and opioids (hypotension, respiratory depression) in percentages ranging from 0.47% to 17% [31]. In this study, all the cases of anesthesia-related complication required an IV bolus infusion of atropine due to persistent bradycardia (HR < 50 bpm) and hypotension (mean pre-sedation level BP value < 50 mmhg). Moreover, flumazenil was administered in one case due to a detected sedation level of five on the Ramsey scale. The aforementioned interventions restabilized the vital parameters and the state of consciousness to normal levels during the drug therapy administration. No invasive and/or non-invasive ventilation, nor advanced resuscitation assistance was needed. None of the patients received opioid antagonists. Remarkably, the statistical analysis of the dataset reported that the increase in the ASA class does not correspond to an increase in anesthesia-related complications. This finding could depend on the personalized drug dosage management, in particular the lowering of the fentanyl dosage, in those who presented more co-morbidities. However, in view of the small number of patients belonging to ASA 3, further studies are needed to confirm this finding. To confirm the safety of sedoanalgesia, the mean value of the Aldrete Score was 9.56, well above the value of 8 that is considered safe for post sedoanalgesia patient discharge. The percentage of endoscopic complications appears very low, having encountered only one post-polypectomy bleeding, halted endoscopically, out of a total of 529 procedures with 139 polypectomies: a complication rate of 0.18% on the total number of procedures and of 0.7% on the total number of polypectomies. The literature reports an incidence of post-polypectomy bleeding in 1.5–2% of cases [32]. We can hypothesize that the deviation from the average levels of complications reported in the literature is due to the sedoanalgesia, to the reduced occurrence of pain-related patient movements, as well as a better working setting for the gastroenterologist. With regard to the efficacy of sedoanalgesia, we encountered a weak point in the study, namely the absence of a satisfaction questionnaire to be filled out by the patient. However, the recorded VAS scores at the end of the procedures clearly show that in 99% of the cases the procedural pain was effectively managed. Another weak point of this study is the absence of a control group; for this reason, especially with regard to the data on the quality of the endoscopic examination, we decided to refer to the literature. In particular, the ADR of 25% appears satisfactory as we know that, according to the ESGE guidelines regarding the quality in screening colonoscopies, an ADR < 20 increases the risk of interval cancer [33]. The cecal intubation rate also appears very satisfactory: 98% for the colonoscopies in the general group, well above the cutoff of 90% indicated by the ESGE guidelines as a goal to be achieved, and 100% in the colorectal cancer screening group, also an optimal result that is superior to the value of 95% identified by ESGE as a target in this specific setting [33]. For this paper, the degree of bowel preparation and the colonoscopy withdrawal time was not assessed, while the ADR and cecal intubation rate were evaluated. As we know from the literature, these two colonoscopy quality indices are related to effective sedoanalgesia during digestive endoscopy procedures [11,13]. The literature states that poor intestinal cleansing is reported in around 30% of colonoscopies [34], and this significantly reduces the ADR and cecal intubation rate [35]. Furthermore, three different types of devices that can be attached to the tip of the colonoscope (Endocuff^TM^; Endoring^TM^; transparent hood or caps) have been shown to increase the ADR [36,37]. However, if we consider that the poor intestinal cleansing rate in the cohort of this study should be consistent with that in the literature data and that no additional connectable devices were used, our results for high ADRs and high cecal intubation rates appear to be equally more relevant. 

### Limitations of the Study

There are several factors that may have influenced the study, though, in our opinion, not the quality of the data. It is an open-label retrospective study without a control group in which the characteristics of patients who underwent sedoanalgesia were analyzed along with the data regarding safety, efficacy and quality of the endoscopic examinations conducted with sedoanalgesia. Therefore, the study has limitations related to its own design: the main ones being the characteristic of a retrospective study and the absence of a control group. For this reason, data obtained were compared with the data already available in the scientific literature. A further limitation is the monocentric nature of the study, which is, however, based on data from a non-university public hospital that serves about 250,000 people and performs about 5000 endoscopic examinations per year. The population pertaining to endoscopy, which was included in the study, counts a not so small number of 529 individuals and has inhomogeneous characteristics, albeit more representative of the real-life population that commonly uses a regular service of digestive endoscopy. This was remedied by stratifying the results on the basis of the ASA class, a factor that, to a greater extent, influences the outcome of the administration of sedative and / or hypnotic drugs [38]. A greater number of endoscopists involved could have perhaps minimized any operator-related bias, but given the purpose of the study, it would have necessarily had to have been endoscopists with similar skill levels. This criterion was respected since our study involved three professionals who were all at the same level from an experience point of view, with at least 10 years of experience in the diagnostic and operative endoscopy field and with documented experience in emergency–urgency and in the use of hypnotic and sedative drugs. 

Given the promising data of this study, it will certainly be advisable to plan a prospective study with a control group in the future. The novelty of the study is that there are good preliminary data for affirming that sedoanalgesia with fentanyl and midazolam-carried out by expert endoscopists is effective and safe. These results are useful to stimulate the design of further randomized and prospective studies to provide strong, clear and shared data. In clinical practice, this knowledge should encourage endoscopists to pay more attention to the issue of procedural pain control and to always approach the use of sedoanalgesia in safe conditions and after careful patient selection. This will improve patient compliance with these procedures, improve the endoscopists work and increase the quality of the examinations. 

## 5. Conclusions

In selected patients, sedoanalgesia managed by the gastroenterologist is very safe and effective when using fentanyl/midazolam with drug dosages that are personalized based on weight, age and ASA score. Moreover, the use of sedoanalgesia in GI endoscopies results in a high-quality endoscopic examination, specifically for cecal intubation rate and ADR and should, therefore, be adopted on a wider scale. 

## Figures and Tables

**Figure 1 jpm-12-01171-f001:**
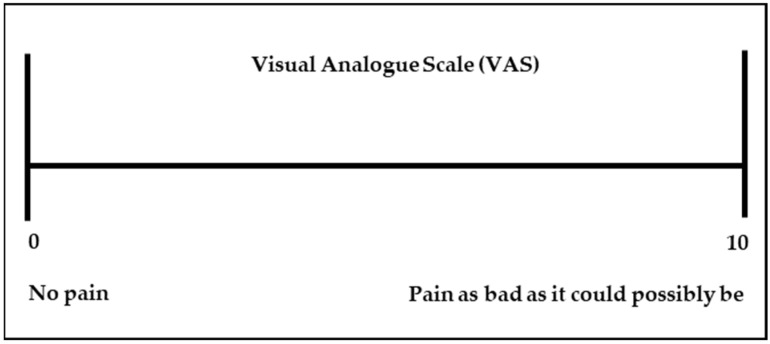
VAS scale.

**Figure 2 jpm-12-01171-f002:**
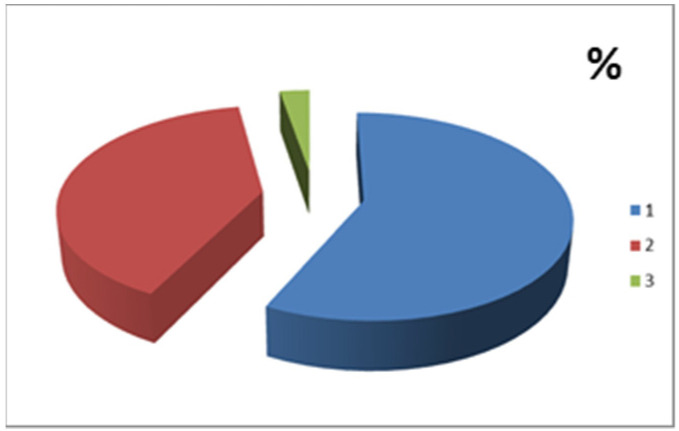
ASA1: 57%, ASA2: 40%, ASA3: 3%.

**Table 1 jpm-12-01171-t001:** ASA class.

ASA Class	Description
I	Healthy individual with no systemic disease
II	Mild systemic disease non-activity limiting
III	Severe systemic disease that limits activity but is not incapacitating
IV	Incapacitating systemic disease that is constantly life threatening
V	Moribund, not expected to survive 24 h with or without operation
VI	A declared brain-dead patient, whose organs are being removed for donor purposes

**Table 2 jpm-12-01171-t002:** Ramsey score.

Sedation Score	Clinical Response
0	Paralyzed, unable to evaluate
1	Awake
2	Lightly sedated
3	Moderately sedated, follows simple commands
4	Deeply sedated, responds to non-painful stimuli
5	Deeply sedated, only responds to painful stimuli
6	Deeply sedated, unresponsive to painful stimuli

**Table 3 jpm-12-01171-t003:** Aldrete score.

Activity	
2	Able to move 4 extremities voluntarily or on command
1	Able to move 2 extremities voluntarily or on command
0	Unable to move extremities voluntarily or on command
Respiration	
2	Able to breathe deeply and cough freely
1	Dyspnea, shallow or limited breathing
0	Apneic
Circulation	
2	BP ± 20 mmhg of pre-sedation level
1	BP ± 20–50 mmhg of pre-sedation level
0	BP ± 50 mmhg of pre sedation level
Consciousness	
2	Fully awake
1	Arousable on calling
0	Not responding
Skin Color	
2	Normal
1	Pale, dusky, blotchy, jaundiced, other
0	Cyanotic

**Table 4 jpm-12-01171-t004:** This table reports the mean of midazolam and fentanyl dosages in the ASA subgroups. The ANOVA test-adjusted by age, sex and weight-was used to test the difference between the means.

	ASA I	ASA II	ASA III	*p*-Value
Mean fentanyl dosage	87.95	79.15	70	<0.001
Mean midazolam dosage	2.12	2.12	1.88	0.306

**Table 5 jpm-12-01171-t005:** Population Characteristic.

Characteristic	N:529
Age Me (IQR)	55 (44, 64)
Gender N (%)	
F	279 (53.8%)
M	250 (46.2%)
Prior surgery N (%)	
N	341 (64.5%)
Y	188 (35.5%)
Weight Me(IQR)	71 (60, 80)
Drug allergy N (%)	
n	490 (92.6%)
y	39 (7.4%)
Pts provenance N (%)	
outpatients	511 (96.6%)
hospitalized	18 (3.4%)
Examination N (%)	
Colonoscopy	443 (83.7%)
EDGS	81 (15.3%)
EDGS + Colonoscopy	5 (1%)

**Table 6 jpm-12-01171-t006:** Cohort data by ASA class *.

Variable	Overall, N = 529	1, N = 303	2, N = 211	3, N = 15	*p*-Value
Diastolic BP pre procedure	77.16 (10.19)	75.87 (10.14)	79.01 (9.99)	77.00 (10.82)	0.003
Diastolic BP post procedure	71.91 (9.74)	70.80 (8.95)	73.57 (10.60)	71.00 (9.67)	0.006
Systolic BP pre procedure	128.63 (19.48)	124.30 (17.80)	134.03 (19.82)	140.00 (24.57)	<0.001
Systolic BP post procedure	116.49 (15.75)	112.78 (13.68)	120.77 (16.09)	131.27 (25.07)	<0.001
HR pre procedure	79.64 (14.19)	80.82 (14.99)	78.33 (12.97)	74.20 (11.68)	0.048
HR post procedure	74.94 (11.86)	75.86 (12.66)	73.89 (10.56)	71.13 (11.22)	0.081
SpO_2_ pre procedure	97.44 (1.83)	97.63 (1.85)	97.17 (1.74)	97.33 (2.32)	0.018
SpO_2_ post procedure	96.31 (2.18)	96.55 (2.13)	95.96 (2.21)	96.40 (2.16)	0.009
Fentanyl dosage					<0.001
25	2 (0.38%)	1 (0.33%)	1 (0.47%)	0 (0.00%)	
50	166 (31.38%)	72 (23.76%)	85 (40.28%)	9 (60.00%)	
75	3 (0.57%)	0 (0.00%)	3 (1.42%)	0 (0.00%)	
100	357 (67.49%)	229 (75.58%)	122 (57.82%)	6 (40.00%)	
125	1 (0.19%)	1 (0.33%)	0 (0.00%)	0 (0.00%)	
Midazolam					0.009
n	6 (1.13%)	3 (0.99%)	1 (0.47%)	2 (13.33%)	
s	523 (98.87%)	300 (99.01%)	210 (99.53%)	13 (86.67%)	
Midazolam dosage					0.050
1.5	70 (13.38%)	40 (13.33%)	25 (11.90%)	5 (38.46%)	
2	291 (55.64%)	164 (54.67%)	120 (57.14%)	7 (53.85%)	
2.5	139 (26.58%)	85 (28.33%)	54 (25.71%)	0 (0.00%)	
3	21 (4.02%)	9 (3.00%)	11 (5.24%)	1 (7.69%)	
3.5	2 (0.38%)	2 (0.67%)	0 (0.00%)	0 (0.00%)	
O_2_ administration					0.4
n	509 (96.22%)	294 (97.03%)	200 (94.79%)	15 (100.00%)	
s	20 (3.78%)	9 (2.97%)	11 (5.21%)	0 (0.00%)	
Aldrete score					0.056
7	1 (0.19%)	0 (0.00%)	0 (0.00%)	1 (6.67%)	
8	34 (6.43%)	16 (5.28%)	17 (8.06%)	1 (6.67%)	
9	159 (30.06%)	87 (28.71%)	69 (32.70%)	3 (20.00%)	
10	335 (63.33%)	200 (66.01%)	125 (59.24%)	10 (66.67%)	
VAS scale					0.056
<4	524 (99.05%)	302 (99.67%)	208 (98.58%)	14 (93.33%)	
>4	5 (0.95%)	1 (0.33%)	3 (1.42%)	1 (6.67%)	
Other drugs					0.7
n	524 (99.05%)	299 (98.68%)	210 (99.53%)	15 (100.00%)	
s	5 (0.95%)	4 (1.32%)	1 (0.47%)	0 (0.00%)	
Anesthesia-related complication					>0.9
n	525 (99.24%)	301 (99.34%)	209 (99.05%)	15 (100.00%)	
s	4 (0.76%)	2 (0.66%)	2 (0.95%)	0 (0.00%)	
Endoscopic complication					
n	529 (100.00%)	303 (100.00%)	211 (100.00%)	15 (100.00%)	

***** The categorical variables are shown with N (%), whereas the continuous variables are shown by Me (IQR).

**Table 7 jpm-12-01171-t007:** Colonoscopy quality data.

Characteristic	N:529
0	336 (75%)
1	83 (19%)
2	15 (3.3%)
3	8 (1.8%)
4	3 (0.7%)
5	2 (0.4%)
7	1 (0.2%)
**Cecal intubation on general population**	
n	10 (2.2%)
y	438 (98%)

## Data Availability

Not applicable.

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
