# Peer review of "Safety, Efficacy and High-Quality Standards of Gastrointestinal Endoscopy Procedures in Personalized Sedoanalgesia Managed by the Gastroenterologist: A Retrospective Study"

_jpm, 2022, doi:10.3390/jpm12071171_

Round 1

Reviewer 1 Report

The manuscript is well written, however, I wonder that until now in Italy Gastroenterologists don't use propofol, which is the common practice in most countries and of course with other quality of sedation compared to midazolam and fentanly, in which the procedure is painful, of course, the lack of the patients experience in the study is a major limitation of the study.  

There are several minor revisions that should be improved:

1. Table 1: please use the updated ASA score, in which the grades 1-6 are used, so please add the score 6 with its definition.

2.  The data in figure 2 are included in table 4, no need to repeat the same results twice. 

Author Response

Dear Reviewer,

thank you so much for your comments and suggestions.

I'm agree with you about the bad habit of Italian Gastroenterologists not to use Propofol for sedation in GI endoscopy, but it's  a problem related to medico- legal issue, due to local uncertain regulation.

In regards to the limitation of the study due to a lack of patient's questionnaire at the end of sedoanalgesia, we have underlined this weak point of the work in the discussion (paragraph 4), with this sentence: "with regard to the efficacy of sedoanalgesia, we encountered a week point in the study, namely the absence of a satisfaction questionnaire to be filled out by the patient".

Reviewer 2 Report

Manuscript entitled Safety, efficacy and high-quality standards of Gastrointestinal Endoscopy Procedures in personalized sedoanalgesia managed by the gastroenterologist: a retrospective study. This manuscript does bring us some new knowledge about personalized sedoanalgesia. While the presentation and writing are poor. The abstract needs further revision, the background and objective of this study are not elaborated well. The structure of this paper looks so weird, there are too many paragraphs which made this paper hard to get to the main point. Figure 1 is not clear. I have found some shade along with some paragraphs, which made the manuscript too unclear to show.

Author Response

Dear Reviewer, thank you for your comments and suggestions.

The abstract has been modified in order to be as clear as possible.

The figure 1 has been replaced by another one more clear and the shade along with same paragraphs has been removed.

We regret for the hardness to get the main point of the article, anyway we have followed the journal's format to compose the structure of the paper  

Round 2

Reviewer 2 Report

Dear editor, thanks for inviting me to review this manuscript again. The authors have revised this paper according to my comments. This paper could be accepted in the present form.     

This manuscript is a resubmission of an earlier submission. The following is a list of the peer review reports and author responses from that submission.

Round 1

Reviewer 1 Report

A study by RIzzi et al. reported that the use of analgesics and sedatives during endoscopy resulted in good examination results. Although results of studies similar to this have been reported in the past, there are several concerns, including the fact that this study did not establish a control group.

1. One important weakness of this study is the lack of controls. Therefore, a more detailed analysis using a pool of cases without seadation within the same institution is needed.

2. Exclusion criteria for this study are not clear.

3. Was this study done with a single physician or multiple physicians? What is the level of training of the physicians? If multiple physicians, are their skill levels comparable?

4. The limitations of this study should be described in detail. First of all, this is a retrospective, single-center study. The authors should also note the possibility of multiple surgeons, different patient populations, etc.

5. Many factors and techniques have been evaluated to improve ADR and PDR, including devices attached to the tip of the colonoscope and withdrawal time, which are important in comparing screening results. The authors should discuss these issues in the text.

Reviewer 2 Report

The study is aimed to verify the safety and efficacy of GI Endoscopy procedural sedoanalgesia, conducted by the gastroenterologist, in selected patients with patient personalized dosages of sedative (Midazolam) and opioid (Fentanyl) pharmaceutical drugs.  The title is “Safety, efficacy and high-quality standards of Gastrointestinal Endoscopy Procedures in personalized sedoanalgesia managed by the gastroenterologist: a retrospective study”.

1.        This is a retrospective study.    Some limitations might have occurred.

2.        Several factors influence the outcome of the study.  Please discuss these.

3.        How did the authors correct some data?

4.        According to the study design, the safety of sedation was not definitely assessed.

5.        Capnometry was not used in this study.  Hypercapnia and/or respiratory depression were not assessed.

6.        What was the definition of sedation-related complications?  How did you evaluate these?

7.        What was the targeted depth of sedation level?   How did the authors assess the depth of sedation level during the procedure?

8.        “The vital parameters were measured pre and post-procedure.”  The sedation-related adverse events did not correct during the endoscopic procedure.

9.        Who were the endoscopists?   How about the endoscopists’ experience?

10.   Please add the limitations of the study?

11.   What is the new knowledge of the study?

12.   Please recommend to the readers “How to apply this knowledge in clinical practice?”.